# Sonocatalytic Activity of Porous Carbonaceous Materials for the Selective Oxidation of 4-Hydroxy-3,5-dimethoxybenzyl Alcohol

**DOI:** 10.3390/molecules29071436

**Published:** 2024-03-23

**Authors:** Behdokht Hashemi Hosseini, Rafael L. Oliveira, Dariusz Łomot, Olga Chernyayeva, Juan C. Colmenares Quintero

**Affiliations:** 1Institute of Physical Chemistry, Polish Academy of Sciences, Marcina Kasprzaka 44/52, 01-224 Warsaw, Poland; dlomot@ichf.edu.pl (D.Ł.); ochernyayeva@ichf.edu.pl (O.C.); 2Institute of Low Temperature and Research Structure, Polish Academy of Science, Okólna 2, 50-422 Wrocław, Poland; r.oliveira@intibs.pl; 3Chemistry Department, Federal University of Pernambuco, Av. Jorn. Anibal Fernandes, Recife 50740-560, PE, Brazil

**Keywords:** sonication, sonocatalysis, selective oxidation, lignocellulosic model compounds, heteroatom-doped carbons, green chemistry, mesoporous carbon

## Abstract

Selective oxidation, which is crucial in diverse chemical industries, transforms harmful chemicals into valuable compounds. Heterogeneous sonocatalysis, an emerging sustainable approach, urges in-depth exploration. In this work, we investigated N-doped or non-doped carbonaceous materials as alternatives to scarce, economically sensitive metal-based catalysts. Having synthesized diverse carbons using a hard-template technique, we subjected them to sonication at frequencies of 22, 100, 500, and 800 kHz with a 50% amplitude. Sonochemical reaction catalytic tests considerably increased the catalytic activity of C-meso (non-doped mesoporous carbon material). The scavenger test showed a radical formation when this catalyst was used. N-doped carbons did not show adequate and consistent sonoactivity for the selective oxidation of 4-Hydroxy-3,5 dimethoxybenzyl alcohol in comparison with control conditions without sonication, which might be associated with an acid–base interaction between the catalysts and the substrate and sonoactivity prohibition by piridinic nitrogen in N-doped catalysts.

## 1. Introduction

The substitution of petroleum-based sources with their renewable counterparts stands as one of the most continuous initiatives in global energy demands. This strategic shift, while crucial, involves complexities, particularly in securing an appropriate carbon source from abundant biopolymers such as cellulose, chitin, alginate, and lignin [1,2,3]. Coniferyl and sinapyl alcohols are two important monomeric units of lignin; thus, the exploration of different strategies in the selective oxidation of these monomers is crucial to produce valuable chemicals [4,5,6,7].

Researchers have been exploring the application of heterogeneous catalysis in selective oxidation reactions to provide a more environmentally friendly alternative to traditional reactions using toxic chemical oxidants. Many strategies, such as thermal, photo, or sonocatalysis, have been reported [8,9,10,11,12,13,14,15,16,17,18].

Ultrasound technology is user-friendly, safe, and produces minimal by-products [19,20,21,22,23,24]. The sonocatalysis mechanism involves cavitation initiated by acoustic ultrasound waves. This process induces periodic impaction expansion, forming micro-cavities or vapor-filled bubbles with substantial growth potential. The extreme conditions within these bubbles, containing trapped energy, lead to explosive bubble collapse, releasing exceptional heat and pressure into the liquid, surpassing 5000 K and 1000 bar [25,26].

Metal-based materials often produce efficient, heterogeneous sonocatalysis, facilitated by the electron transfer mechanism. It is noteworthy to highlight that the extent of this phenomenon depends on system-specific characteristics, including surface tension, density, temperature, and the nature of the phases involved [27]. The drawback of metal-based catalysts is the elevated cost associated with mining methods to extract metals and their purification. Metal-free catalysts have steered research towards cheaper, more abundant alternatives, such as carbon materials. This burgeoning class of materials in catalysis facilitates the investigation of novel catalytic processes in the absence of metals [28,29,30,31,32].

Carbon-based materials have been extensively studied as sonocatalysts [33,34,35]. They exhibit both nucleation effects and unique physicochemical properties that enhance the sonocatalytic degradation process [36,37]. A range of carbonaceous materials, including carbon nanotubes (CNTs), graphene (GR), graphene oxide (GO), reduced graphene oxide (rGO), activated carbon (AC), biochar (BC), g-C3N4, buckminsterfullerene (C60), mesoporous carbon, and doped-carbon have garnered attention for water and wastewater treatment applications and catalysis [17,38,39,40].

The incorporation of heteroatoms such as nitrogen, phosphorus, boron, and/or sulfur modifies the physicochemical characteristics of carbon materials. Heteroatom-doped carbons have emerged as a captivating material category in catalysis, offering avenues for investigating novel metal-free catalytic processes and catalyst supports. Depending on their chemical arrangement within the carbon matrix, heteroatoms can function as either Lewis acids or Lewis bases, exhibiting diverse roles in catalytic processes [28].

No specific study has been conducted on the carbonaceous, heterogeneous, sonocatalytic selective oxidation of lignin model compounds. Therefore, there is an interest in evaluating the effect of N-doping. In this work, mesoporous carbon materials (N-doped and non-doped) were synthesized and examined as sonocatalysts or thermal catalysts for the selective oxidation of 4-hydroxy-3,5-dimethoxybenzyl alcohol. The influence of the dopant and the chemical composition of carbon materials were explored. Moreover, we carried out experiments to test the catalysts at different frequencies, aiming to understand the essential factors that influence the selective oxidation of alcohols. Calorimetry was conducted to precisely measure the energy requirements for different experiments at various frequencies, i.e., 22, 100, 500, and 800 kHz, all with a consistent amplitude of 50%. In control conditions without sonication, pyridinic N was recognized as a vital site on N-doped materials as an acid–base link point. Under sonocatalysis conditions, non-doped carbon was the most efficient material due to radical formations.

## 2. Results and Discussion

### 2.1. Characterization

Figure 1a shows an illustrative scheme of the porous synthesis of carbonaceous materials. The hard-template technique was used to synthesize four different materials using distinct carbon and nitrogen sources: sucrose (C-meso), chitosan (CN-meso chit), glucosamine (CN-meso Gluc), and a mixture of sucrose and cyanamide (CN-meso). Figure 1b shows the N_2_ physisorption isotherm of the synthesized materials. The isotherm shows a unique capillary condensation step in the range of 0.7–0.95 in the absorption branch, indicating the occupation of N_2_ molecules into the uniform mesopores. Following the IUPAC classification, the isotherms can be classified as type IV. Table 1 shows the BET surface area and pore volume of the obtained materials. C-meso presents a slightly higher surface area than N-doped carbon materials.

The XRD patterns of the materials are displayed in Figure 1c. Two broad peaks are observed at 25° and 45°; these reflections correspond to (002) and (100) lattice planes, demonstrating a poorly ordered graphitic structure. Raman spectra of the carbonaceous materials are shown in Figure 1d. Two bands are observed at 1355 cm^−1^ and 1590 cm^−1^, corresponding to the D-band and G-band. The D-band is related to imperfections on sites in the sp^2^ carbon lattice [41]. The ratio of the areas (ID/IG) is associated with the degree of disorder and imperfections in these materials. The values are presented in Table 1. Among the carbonaceous materials, CN-meso presents a structure with the most imperfections.

Figure 2a–d show SEM images of the carbonaceous materials. As expected, the materials presented random morphologies with different particle sizes. This is typically observed in materials synthesized using the hard-template methodology. On the other hand, the TEM image in Figure 2e–h shows materials with a very uniform pore size in the range of silica nanoparticles.

Fourier transform infrared (FTIR) spectra of carbonaceous materials are shown in the Appendix A. A weak and broad band at 3500 cm^−1^ is shown, assigned to the stretching vibrations of O to endash O–H of carboxylic acid or absorbed water. And one more intense peak is displayed at around 1710 cm^−1^ and corresponds to C=O stretching vibrations in carbonyl or carboxyl groups. A band at around 1500 cm^−1^ is normally associated with C=C or C=N bonds. The peaks at 925 cm^−1^ and 770 cm^−1^ are related to the bending of C=C and C–H, respectively [42].

TG analyses are shown in the Appendix A. All the carbonaceous materials faced a considerable mass loss between 400 °C and 600 °C due to carbon structure combustion, and a further mass loss at 800 °C followed this step. All materials lost almost their complete mass, showing that the silica template was almost completely dissolved in the sodium hydroxide solution.

The chemical compositions extracted from XPS analyses of the synthesized carbonaceous materials and the high-resolution XPS spectra of C1s are shown in Figure 3. The XPS C1s spectrum was deconvoluted into six distinguished peaks at 284.2, 258.9, 287.5, 289.2, 290.7, and 292.5 eV. These peaks are associated with C sp^2^, C sp^3^, and carbon bound to oxygen (C–O) in esters or phenol structures, the double bond between carbon and oxygen (C=O) in quinones and ketones, two oxygen atoms in carboxylic acid, carboxyl, or esters (–COO), and carbonates, respectively [28,29,30,31,32,33,34,35,36,37,38,39,40,41,42,43]. It is interesting to point out that the carbonaceous materials derived from sucrose (C-meso and CN-meso) present a smaller ratio between Csp^3^ and Csp^2^ than CN-meso Chit and CN-meso Gluc. Liu et al. associated this ratio between Csp^3^ and Csp^2^ in the XPS analyses with more defective carbon production. Their observation is in line with the data reported herein, where the Raman spectra show that C-meso and CN-meso present higher ID/IG values than the other N-doped carbon reported herein [44]. These defective structures have been shown to favor some electrochemical and thermal catalytic activities.

The percentage of oxygenated species on the carbon surface considerably changed depending on the chemical precursors used. The carbonaceous materials derived from sucrose (C-meso and CN-meso) present 12 and 10%, in contrast with CN-meso Chit and CN-meso Gluc, which displayed a much lower concentration of this functional group, i.e., 4 and 5%, respectively. These oxygenated groups made the carbonaceous materials more hydrophilic, which can be extremely useful for running reactions in aqueous solutions. Moreover, these groups could also act as electron donors. The high-resolution O1s XPS was split into four peaks centered at 531.0, 532.9, 535.3, and 537.8 eV (Appendix A). These peaks correspond to C=O, O–C=O, C–O–H, and adsorbed H_2_O or C–O [45]. The distribution of oxygenated species also changes with the selection of carbon precursors.

The high-resolution N1s XPS was used to picture the position that nitrogen species occupied in the carbon frame (Figure 4). The spectra could be deconvoluted into three or four peaks centered at 398.1, 400.2, 401.9, and 404.3 eV, which correspond to N-pyridinic, N-pyrrolic, N-graphitic, and Noxide, respectively. In all cases, N-pyridinic and N-pyrrolic are present as the major species. The pyridinic type is located at the edges or in vacancies; the lone pair of N-pyridinic gives the N-doped carbon a Lewis basicity. Moreover, N-pyridinic can act as an adsorption point when slightly acid substrates are used as adsorbates or substrates in catalytic processes [46].

### 2.2. Catalytic Performance

The selective oxidation of 4-hydroxy-3,5-dimethoxybenzyl alcohol was selected as a model reaction. The results from the control conditions without sonication (conversions and selectivity to the corresponding aldehyde) are shown in Figure 5A. N-doped carbons show better conversion and selectivity to aldehyde in a control condition without a sonication catalytic test than non-doped carbon (C-meso), which showed no catalytic activity. Among N-doped carbons, CN-meso Chit and CN-meso Gluc show less effective selectivity. It is known that N-pyridinic can act as an adsorption point to —OH groups from alcohols, which commonly present weak acidity properties. Moreover, some researchers have suggested that N-pyridinic can cause the deprotonation of alcohols, which is known to be a rate-determined step [47]. Thus, the concentration and distribution of these N-species can play an important role in conversion and selectivity due to contact time and the equilibrium between adsorption and desorption.

CN-meso shows the best performance, with high conversion and a good selectivity to the corresponding aldehyde. As shown by XPS spectroscopy, CN-meso presents the lowest percentage of N combined with a highly defective structure. A lower percentage of nitrogen would decrease the adsorption point and decrease the interaction time between the substrate and catalyst when compared to other N-doped carbons with a high percentage of N-species. This might explain the superior selectivity performance observed for CN-meso, as further oxidation steps (aldehyde to carboxylic acid) were less pronounced.

It is important to point out that defects in the carbon frame can also influence the catalytic activity of these materials. Cheng et al. also related the degree of defects with the formation of singlet oxygen (^1^O_2_), a non-radical reaction path mechanism, in the total oxidation of 2,4 dichlorophenol using carbon nanofibers and PDS. Moreover, a high percentage of oxygenated species present in CN-meso are known to enhance the hydrophilicity of this carbonaceous structure, which might also influence the catalytic activity [48].

Figure 5B shows the conversion and selectivity of selective oxidation of 4-hydroxy-3,5-dimethoxybenzyl alcohol under sonocatalysis conditions. Without the presence of carbonaceous materials, no conversion of substrate was observed. Among N-doped carbons, CN-meso Chit and CN-meso Gluc faced a slight decrease in conversion values and a considerable decrease in selectivity. In the case of CN-meso, conversion dropped considerably. The presence of ultrasound might have interfered with the interaction between the substrate and the catalyst, as the solid presents a lower percentage of N (fewer adsorption points), which results in a much lower catalytic activity than other N-doped carbonaceous materials.

C-meso improved its performance considerably under sonocatalysis (Figure 5B). C-meso was used as the catalyst for further investigations of sonochemical reaction catalytic tests. These experiments were performed at different frequencies: 22, 100, 500, and 800 kHz. They are summarized in Figure 6. The best sonocatalytic performances were at 22 kHz. However, despite achieving high conversion rates, this material did not exhibit selectivity towards our desired products, the corresponding aldehyde and carboxylic acid (syringaldehyde and syringic acid).

It is worth mentioning that, although we observed adequate conversion in control conditions without sonication for the N-doped catalyst, we decided to apply the same frequencies of sonication (22, 100, 500, and 800 kHz). As we show in the Appendix A, fluctuations in conversion rates stem from the interaction of mechanical effects and oxidative agents during sonication. Higher frequencies emphasize oxidative species, while lower frequencies involve both mechanical and oxidative effects. Over time, the system stabilizes, indicating a more consistent impact of sonication. Not being reproducible is another problem in N-doped heterogenous sonochemical reaction catalytic tests.

Numerous control tests, encompassing experiments with and without sonication, were conducted, along with sonolysis experiments of 4-hydroxy-3,5-dimethoxybenzyl alcohol at varying frequencies. Interestingly, no sonolysis was observed for the aforementioned substrate.

Furthermore, we conducted several sonochemical reaction catalytic tests by substituting the solvent with acetonitrile. However, these tests yielded no activity. We attribute this lack of activity to the cavitation phenomenon, alongside the presence of water and the hydrophilicity of our catalyst.

In addition to the solvent change, we explored the substrate’s reaction with a widely recognized carbonaceous catalyst, g-C^3^N^4^. Surprisingly, in all of these catalyst tests, we observed neither conversion nor selectivity.

To delve deeper into the mechanisms behind N-doped and non-N-doped catalysts, we conducted scavenger tests. We utilized ethanol as a •OH scavenger, employing ratios of 1:1 and 1:2 with the catalyst in a 1 mM solution of 4-Hydroxy-3,5-dimethoxybenzyl alcohol. Remarkably, in the case of C-meso catalysts, the scavenger hindered the reaction. Conversely, all other catalysts exhibited continuous reactions. This suggests a notable advancement in non-radical reactions under sonication when utilizing N-doped catalysts. Figure 7 depict how the reaction ceased in the sonochemical reaction test after adding the scavenger after 1 h, as the reaction was progressing with the addition of ethanol, and how it persisted in the case of N-doped catalysts.

## 3. Materials and Methods

### 3.1. Experimental

Materials for synthesis and reagents for sonochemical reaction catalytic tests (set up scheme shown in Figure 1) with 4-hydroxy-3,4-dimethoxybenzylalcohol were purchased from ThermoFisher (Waltham, MA, USA). LUDOX HS-40 colloidal silica (40 wt% suspension in water) and cyanamide (99%) were purchased from Sigma Aldrich (St. Louis, MO, USA). Sucrose and sodium hydroxide were purchased from POCH (Gliwice, Poland). D-glucosamine hydrochloride (≥98%) and low-molecular-weight chitosan (100–300 cps) were purchased from Pol-Aura. Acetic acid (99.5%) was purchased from ChemPur (Karlsruhe, Germany). Milli-Q water was purified using a Millipore Milli-Q lab water system (Burlington, MA, USA). The carbonaceous materials were characterized by nitrogen physisorption, thermogravimetric analysis (TGA), X-ray photoelectron spectroscopy (XPS), scanning electron microscopy (SEM), and Fourier transform infrared spectroscopy (FTIR). The equipment used and details of the analysis are shown in the Appendix A.

#### Synthetic Methods for Porous Heteroatom-Doped Carbons

**C-meso (non-doped catalyst):** Sucrose (2.0 g) was dissolved in water (10 mL). After dissolving the sugar, 10 g of colloidal silica (LUDOX HS-40) was slowly added dropwise. After mixing for 10 min, the solvent was removed under a vacuum at 60 °C. The obtained paste was placed into a tubular oven for thermal treatment (heating to 800 °C at a rate of 3 °C·min^−1^ followed by annealing at 800 °C for 1 h) under argon flow. The obtained black powder was added to a 6 M NaOH solution and stirred for 48 h. The obtained solid was filtered and washed extensively with hot water (60 °C).

**CN-meso:** This carbonaceous material was synthesized using a modified recently described methodology [39]. In this case, 1.5 g of sucrose and 0.5 g of cyanamide were dissolved in 10 mL of water. Then, 10 g of colloidal silica solution (LUDOX HS-40) was added to this solution and mixed for 10 min. The solution was dried under a vacuum at 60 °C. The obtained paste was placed into a tubular oven for thermal treatment (heating to 800 °C at a rate of 3 °C·min^−1^ followed by annealing at 800 °C for 1 h) under argon flow. The removal of the template was performed as described for C-meso.

**CN-meso Chit:** This material was synthesized using a previously developed methodology [18]. In summary, acetic acid (4 mL) was added to 100 mL of water; then, 2 g of chitosan was added. This mixture was stirred overnight to form a hydrogel. Then, 10 g of colloidal silica solution (LUDOX HS-40) was introduced dropwise into the formed hydrogel. This solution was dried at 40 °C until the formation of a dark yellowish film. This film was placed into a tubular oven for thermal treatment (heating to 800 °C at a rate of 3 °C·min^−1^ followed by annealing at 800 °C for 1 h) under argon flow. The silica removal was performed following the same procedure mentioned for C-meso.

**CN-meso Gluc:** This N-doped carbon was synthesized using glucosamine as the C and N precursor. A total of 2 g of glucosamine was dissolved in 10 mL of water. Following the addition of 10 g of colloidal silica (LUDOX HS-40), the solvent was removed by vacuum at 60 °C. The obtained white powder was placed into a tubular oven for thermal treatment (heating to 800 °C at a rate of 3 °C·min^−1^ followed by annealing at 800 °C for 1 h) under argon flow. The obtained black powder was added to a 6M NaOH solution (50 mL) and stirred for 48 h, aiming at silica removal. The obtained solid was filtered and washed extensively with hot water (60 °C, approximately 700 mL of water).

## 4. Conclusions

Mesoporous N-doped carbon materials, both doped and non-doped, were synthesized and evaluated as sonocatalysts or thermal catalysts for the selectively oxidation of 4-hydroxy-3,5-dimethoxybenzyl alcohol. The impact of dopants and the chemical composition of the carbon materials were investigated, including experiments at different frequencies to unravel factors influencing alcohol oxidation.

The percentage of oxygenated species on the carbon surface varied with precursors, impacting hydrophilicity. High-resolution XPS analyses unveiled peaks corresponding to different oxygenated and nitrogen species. N-pyridinic’s nature significantly influenced alcohol adsorption, affecting conversion and selectivity, which led to a better performance of N-doped carbon materials in control conditions without sonication. This adsorption phenomenon might be disturbed under sonication; thus, N-doped carbons present a decrease in activity and selectivity. Overall, this exploration underscores the potential of N-doped carbonaceous catalysts in heterogeneous sonocatalysis for selective alcohol oxidation, shedding light on the vital role of specific nitrogen species and carbon composition in catalytic activity.

Future investigations will encompass two primary aspects: structural analysis and a sonochemical reaction catalytic test. We will focus on various carbon structures in the sonocatalysis. Additionally, we aim to conduct sonochemical reaction catalytic tests using the same catalysts within a flow system. These dual approaches will provide comprehensive insights into the efficacy and mechanisms underlying carbon-based sonocatalysts.

## Data Availability

Data are contained within the article and Appendix A.

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
