# Peer review of "Sonocatalytic Activity of Porous Carbonaceous Materials for the Selective Oxidation of 4-Hydroxy-3,5-dimethoxybenzyl Alcohol"

_molecules, 2024, doi:10.3390/molecules29071436_

Round 1

Reviewer 1 Report

Comments and Suggestions for Authors

The manuscript titled "Sonocatalytic Activity of Porous N-Doped Carbonaceous Materials for the Selective Oxidation of 4-hydroxy-3,5-dimethoxybenzyl Alcohol" presents a topic of considerable interest that is closely aligned with current hotspots in scientific research. The work proposed on the catalytic role of N-doped porous carbon materials in specific oxidation reactions represents an area worthy of exploration, which is significant for understanding and enhancing catalytic efficiency. The article has some areas that need improvement and a minor  revision is recommended.

1. I recommend that the entire document be polished for language. The sentence from the abstract is lengthy and complex, which may lead to confusion. It also appears to contain several grammatical and stylistic issues. Here is a detailed breakdown of these problems:

1. **Length and Complexity:**

   - The sentence is quite lengthy, which can make it difficult for readers to follow the flow of information.

2. **Grammatical Issues:**

   - The phrase "We aimed to identify the most effective sonocatalytic effect of N-doped carbons show" is grammatically incorrect. It seems to be missing a verb for the clause "N-doped carbons show" and could be better structured.

   - The phrase "under silent controlling tests" is unclear. It might be better to say "under control conditions without sonication."

   - The terms "silent controlling tests" and "sonocatalytic tests" are not standard and could confuse readers. Standard terminology should be used to ensure clarity.

3. **Punctuation:**

   - The sentence lacks appropriate punctuation, which contributes to its confusing structure.

4. **Use of Jargon:**

   - The use of technical terms like "acid-based interaction," "C-meso," and "pyridinic nitrogen" without proper explanation can make the sentence difficult for readers not specialized in the field to understand.

2. The introduction should appropriately cite specific researchers' progress in this direction and the relevant performance data.

3. There are some formatting issues within the text; please thoroughly proofread and revise.

**Formatting issues:**

The first letter of the title of the table is not capitalised. The thickness of the horizontal lines in the tables is not uniform.

There is an error in the figure labels' sequence at line 146; please check and correct throughout the document. Additionally, please label the peak positions in the XPS spectra for the convenience of the readers.

4. The conclusion drawn from the investigation presents a nuanced understanding of the role of N-doping in enhancing the sonocatalytic oxidation of alcohols. However, there are certain limitations and areas where further experimental evidence is required:

1. **Lack of Comparative Analysis:** While the study indicates that N-doped carbon materials outperform their non-doped counterparts, it lacks a detailed comparative analysis that quantifies the improvement. A benchmark comparison with industry-standard materials could provide a clearer picture of the performance enhancement due to N-doping.

2. **Surface Oxygenated Species:** The conclusion suggests that the amount of oxygenated species on the carbon surface varies with precursors, but it does not provide a clear correlation between the quantity of these species and the catalytic performance. Quantitative analysis in this area could strengthen the understanding of this relationship.

3. **High-resolution XPS Analysis:** Although high-resolution XPS analyses have unveiled peaks corresponding to different oxygenated and nitrogen species, the study does not fully explore the link between these surface species and the observed catalytic activity. A more in-depth analysis could help in identifying the active sites responsible for catalysis.

4. **Alcohol Adsorption Mechanisms:** The nature of N-pyridinic is highlighted as a significant factor influencing alcohol adsorption, yet the mechanisms through which this occurs are not elaborated upon. Experimental data on adsorption isotherms and kinetics would be valuable in elucidating these mechanisms.

5. **Influence of Nitrogen Species:** The study underscores the vital role of specific nitrogen species and carbon composition in catalytic activity but falls short of providing a detailed mechanism of how these nitrogen species affect the conversion and selectivity. Isotopic labeling experiments and in-situ spectroscopy could provide direct evidence of the reaction pathway.

6. **Electrophilic Pyridinic Nitrogen:** The role of electrophilic pyridinic nitrogen in prohibiting the attenuation of sonoactivity due to N-doping is mentioned, but the study does not explain why this is the case. Computational modeling or quantum chemical calculations could offer insights into the electronic effects induced by N-doping.

7. **Synergies between Functional Groups:** The interaction between pyridinic N and carbonyl C=O is recognized as important, but a detailed study on how these functional groups work in tandem during sonocatalysis is missing. In situ spectroscopic methods could be used to monitor these interactions during the reaction.

In future work, it is suggested to address these limitations by:

- Conducting a comparative study with standard materials.

- Quantifying the role of surface oxygenated species in catalytic performance.

- Providing a detailed mechanism of alcohol adsorption and the influence of nitrogen species.

- Exploring the electronic structure changes due to N-doping and their effect on sonoactivity.

- Investigating the synergistic effect between pyridinic N and carbonyl C=O groups in greater detail.

The proposed future investigations focusing on structural analysis and sonocatalytic assessments, along with tests within a flow system, are promising. However, integrating the suggestions above could significantly contribute to the field by providing a deeper understanding of the mechanisms at play in N-doped carbon-based sonocatalysis.

5. You may consider referring to the following articles for additional insights and perspectives:

1.   https://doi.org/10.1016/j.ceramint.2024.02.135

2.   10.3390/nano13101664

Comments on the Quality of English Language

I recommend that the entire document be polished for language. The sentence from the abstract is lengthy and complex, which may lead to confusion. It also appears to contain several grammatical and stylistic issues. 

Author Response

Thank you for your dedicated attention and time spent on reviewing the material. We have addressed all the questions, and you may find the answers enclosed in the attached document.

Reviewer 2 Report

Comments and Suggestions for Authors

This manuscript reports the synthesis of porous nitrogen-doped carbonaceous materials and their sonocatalytic effects for the selective oxidation of 4-Hydroxy-3,5 dimethoxybenzyl alcohol, a lignin-derived phenolic monomer with the aim of producing valuable sustainable chemicals.

Questions:

1.    For the N2 physisorption isotherms for the carbonaceous materials in Figure 1(b), the trends of the quantity adsorbed (cm3/g STP) values are inconsistent with the surface areas and pore volumes on Table 1. For example, even though C-meso has a larger surface area and pore volume in comparison to CN-meso, based on Figure 1(b), C-meso shows a lower quantity adsorbed in comparison to CN-meso. Can you elaborate on this discrepancy?

2.    On page 5, Figure 3 was mistakenly labelled as Figure 1.

3.    On page 6, Figure 5 was mistakenly labelled as Figure 2.

4.    The structure used for 4-hydroxy-3,5-dimethoxybenzyl alcohol on Figure 5 and 6 is not the correct structure.

5.    The results for the conversion and selectivity in the selective oxidation of 4-hydroxy-3,5-dimethoxybenzyl alcohol under sonocatalytic tests for the CN-based carbonaceous materials performed at different frequencies: 22 kHz, 100 kHz, 500 kHz, 209 and 800 kHz shown in the supporting information on Figure 3–5, were not discussed in the manuscript. This information is critical and should be discussed in the manuscript.  

6.    The results for the scavenger test in the supporting information should also be included as a discussion in the manuscript. The test results should be used to explain the sonocatalytic activity of 4-hydroxy-3,5-dimethoxybenzyl alcohol for the catalysts. How does the scavenging ability affect conversion and selectivity for the substrate with and without sonication?

Comments on the Quality of English Language

Improvements can be made for the results and discussion section. Some of the figures were not properly labelled. Also, there's some level of incoherency with the discussion. 

Author Response

(The authors gave the same response as above.)

Reviewer 3 Report

Comments and Suggestions for Authors

I think this work is well, I only suggest the authors give a sub-head for comparing the current work and documents.

Author Response

(The authors gave the same response as above.)

Round 2

Reviewer 1 Report

Comments and Suggestions for Authors

Please check Figure 1 labelling. There is a duplication of the labelling of figure 1.